# Samarium-Doped Lead Magnesium Niobate-Lead Titanate Ceramics Fabricated by Sintering the Mixture of Two Different Crystalline Phases

**DOI:** 10.3390/ma16206781

**Published:** 2023-10-20

**Authors:** Guo-Cui Bao, Dong-Liang Shi, Jia-Ming Zhang, Fan Yang, Guang Yang, Kun Li, Bi-Jun Fang, Kwok-Ho Lam

**Affiliations:** 1Department of Electrical and Electronic Engineering, The Hong Kong Polytechnic University, Hong Kong, China; guocbao@polyu.edu.hk (G.-C.B.);; 2Department of Applied Physics, The Hong Kong Polytechnic University, Hong Kong, China; 3School of Petrochemical Engineering, Changzhou University, Changzhou 213164, China; 4School of Materials Science and Engineering, Changzhou University, Changzhou 213164, China; 5Centre for Medical and Industrial Ultrasonics, James Watt School of Engineering, University of Glasgow, Glasgow G12 8QQ, UK

**Keywords:** two different crystalline phases, PMNT-PSMNT, piezoelectric properties, dielectric properties

## Abstract

The fabrication method plays a key role in the performance of lead magnesium niobate–lead titanate-based ceramics. (1 − *w*)[Pb(Mg_1/3_Nb_2/3_)_0.67_Ti_0.33_O_3_]-*w*[Pb_1−1.5*x*_Sm*_x_*(Mg_1/3_Nb_2/3_)*_y_*Ti_1−*y*_O_3_] piezoelectric ceramics were prepared by sintering the mixture of two different crystalline phases in which two pre-sintered precursor powders were mixed and co-fired at designated ratios (*w* = 0.3, 0.4, 0.5, 0.6). The X-ray diffraction results show that all the ceramics presented a pure perovskite structure. The grains were closely packed and the average size was ~5.18 μm based on observations from scanning electron microscopy images, making the ceramics have a high density that is 97.8% of the theoretical one. The piezoelectric, dielectric, and ferroelectric properties of the ceramics were investigated systematically. It was found that the properties of the ceramics were significantly enhanced when compared to the ceramics fabricated using the conventional one-step approach. An outstanding piezoelectric coefficient *d*_33_ of 1103 pC/N and relative dielectric permittivity *ε*_33_/*ε*_0_ of 9154 was achieved for the ceramics with *w* = 0.5.

## 1. Introduction

Piezoelectric ceramics are functional ceramics that can transform mechanical energy and electrical energy into each other. Perovskite (1 − *x*)Pb(Mg_1/3_Nb_2/3_)O_3_-*x*PbTiO_3_ (PMN-PT), one of the typical ferroelectrics among the perovskite-type ferroelectrics, tungsten bronze type ferroelectrics, bismuth-layer-type ferroelectrics, lithium niobate-type ferroelectrics, and pyrochlore-type ferroelectrics, features excellent piezoelectric, dielectric, pyroelectric, and electrostrictive properties [1] due to its unique crystal structure. The dipole alignment caused by the movement of A-site and B-site ions plays a critical role in ferroelectricity. In detail, the A-site ions are occupied by Pb^2+^ ions and are located at eight vertices of the cube, while the B-site ions are disorderly dominated by Mg^2+^, Nb^5+^, and Ti^4+^ in the center of the cube, presenting a disordered distribution macroscopically [2]. The mechanism of relaxor ferroelectrics can be explained as follows: (1) Due to the difference in ionic radius and the repulsion against the adjacent electron clouds, the arrangement of ions is loosely bonded. (2) Under an external electric field, the disordered B-site lattice favors larger ions such that smaller ions retain more free space, assisting the movement in oxygen octahedrons [3,4,5].

The fabrication method plays a key role in the performance of ceramics. The common issue of preparing PMN-PT-based materials using the conventional one-step mixed oxide sintering method is the formation of the pyrochlore phase, which significantly deteriorates the properties of the PMN-PT-based ceramics [6,7,8]. Various preparation methods have been studied to avoid the appearance of the pyrochlore phase and to improve the properties of PMN-PT-based ceramics. Guha et al. prepared 0.92PMN-0.08PT ceramics by the excess PbO method, avoiding the pyrochlore phase due to lead oxide volatilization [9]. Liou obtained PMN-PT ceramics with pure dense perovskite structures by the reaction-sintering process [10]. Jayasingh et al. eliminated the formation of the pyrochlore phase and improved the homogeneity of the precursor of PMN-PT ceramics by using a partial oxalate process route [11]. Kong et al. synthesized single-phase perovskite PMN-PT nano-powders via a high-energy ball milling process [12]. Chu et al. found that the particle-coating method was a low-cost and simple process to prepare pyrochlore-free PMN-PT powders [13]. Ghasemifard et al. obtained pyrochlore-free 0.65PMN-0.35PT nano-powders through the auto-combustion method [14]. Ravindranathan et al. developed the sol–gel method to prepare 0.9PMN-0.1PT powders with an almost 100% pure perovskite phase [15].

Doping modification is one of the crucial ways to improve the performance of PMN-PT ceramics. Kobor et al. studied the dielectric and mechanical behaviors of 1%Mn-0.65PMN-0.35PT ceramics and found the Mn^2+^-doped ceramics offered a reduction in dielectric permittivity (3800 to 2074) and an enhanced mechanical quality factor (78 to 317) [16]. Li et al. investigated the relaxor behavior of CuO-0.94 PMN-0.06 PT ceramics and revealed a strongly correlation between phase transition dispersity and B-site order degree [17]. Recently, Li et al. reported a PMN-PT ceramic with ultrahigh piezoelectric coefficients (*d*_33_) of up to 1500 pC/N and dielectric permittivity (*ε*_33_/*ε*_0_) of above 13,000 via the doping of samarium [18]. Due to the unexpectedly high piezoelectric response in Pb-based ferroelectrics, rare earth element doping modification has recently attracted much attention in modifying piezoelectric ceramics [19,20,21,22,23,24].

It is generally recognized that ceramics exhibit outstanding piezoelectric properties at the morphotropic phase boundary (MPB) [25]. For PMN-PT ceramics, when the PT content reaches 33–35% in moles, the crystal structure is a mixture of the rhombohedral phase and tetragonal phase in the vicinity of phase boundaries, denoting the MPB. However, it is not possible to precisely control the ratio of the two crystalline phases at the MPB. Moreover, the pyrochlore phase could be easily induced by the doping of rare earth elements in the PMN-PT system [19,26] due to the high melting point of Sm_2_O_3_ [27] and the poor diffusion ability of Sm^3+^ at high temperatures. In this study, sintering the mixture of two different crystalline phases is proposed to prepare (1 − *w*)PMNT-*w*PSMNT (PMNT-PSMNT) ceramics with the same chemical composition but in different mixing ratios, *w*. The major merits of this newly proposed method are the accurate control of the crystal phase proportion and the reduction or even prevention of the formation of the pyrochlore, resulting in the enhancement of the piezoelectric and dielectric properties. It was found that the overall performance of the PMNT-PSMNT ceramics prepared using the proposed method was significantly improved when compared to those prepared by the conventional method.

## 2. Preparation and Characterizations

### 2.1. Ceramic Preparation

The target formula composition Pb_0.9625_Sm_0.025_(Mg_1/3_Nb_2/3_)_0.72_Ti_0.28_O_3_ was determined. Using the lever principle in the phase equilibrium calculation method, the chemical components on both sides of the target formula Ti = 0.28 were selected, and one component was doped with Sm. Two components with different crystal phases were pre-synthesized and then mixed, formed, and sintered according to the design proportion to prepare the non-uniform doped ceramic system. To avoid the formation of the pyrochlore phase, niobium and titanium compounds (Nb_2_O_5_, TiO_2_) with poor sintering activity were mixed with strong sintering active Mg_2_(OH)_2_CO_3_, and a small amount of lead oxide was used as a diffusion promoter to synthesize two kinds of pre-sintered powders in advance. Analytical-grade chemicals, Pb_3_O_4_ (Aladdin, 99.95% purity), Sm_2_O_3_ (Aladdin, 99.99% purity), TiO_2_ (Aladdin, 99.99% purity), Mg_2_(OH)_2_CO_3_ (Aladdin, 99.99% purity), and Nb_2_O_5_ (Aladdin, 99.99% purity), were used as raw materials to synthesize Pb(Mg_1/3_Nb_2/3_)_0.67_Ti_0.33_O_3_ (Component A) and Pb_1−1.5*x*_Sm*_x_*(Mg_1/3_Nb_2/3_)*_y_*Ti_1−*y*_O_3_ (Component B, B_1_: *x*_1_ = 0.0833, *y*_1_ = 0.8367; B_2_: *x*_2_ = 0.0625, *y*_2_ = 0.7950; B_3_, *x*_3_ = 0.0500, *y*_3_ = 0.7700; B_4_: *x*_4_ = 0.0417, *y*_4_ = 0.7533) ceramic powders. In Component B, Mg_2_(OH)_2_CO_3_, Nb_2_O_5_, Sm_2_O_3_, and TiO_2_ were evenly ground and pre-sintered at 880 °C for 4 h, then crushed and Pb_3_O_4_ was added. Pb_0.9625_Sm_0.025_(Mg_1/3_Nb_2/3_)_0.72_Ti_0.28_O_3_ piezoelectric ceramics with the nominal composition were prepared by mixing Component A and Component B according to the chemical formula of (1 − 0.025*/x*)[Pb(Mg_1/3_Nb_2/3_)_0.67_Ti_0.33_O_3_]-0.025*/x*[Pb_1−1.5*x*_Sm*_x_*(Mg_1/3_Nb_2/3_)*_y_*Ti_1−*y*_O_3_] (0.025*/x* = 0.3, 0.4, 0.5, 0.6), followed by ball milling, granulation, molding, and sintering processes. The preparation process is shown in Figure 1. To simplify the analysis, we denote *w* as 0.025/*x*. The sintering conditions were 1250 °C for 1.5 h. After grinding and polishing, the ceramics were coated with silver paste on both surfaces, and sintered at 650 °C for 30 min to form silver electrodes. The ceramic discs were poled at 3 kV/mm for 1 min in silicone oil at room temperature, aged and short-circuited for 24 h, and then the electrical properties were measured. For comparison, Pb_0.9625_Sm_0.025_(Mg_1/3_Nb_2/3_)_0.72_Ti_0.28_O_3_ was prepared by the traditional mixed oxide sintering process as a reference sample.

### 2.2. Characterizations

D/max-2500/PC X-ray powder diffraction (XRD, Rigaku, Tokyo, Japan) with Cu K_α_ radiation and JSM-6510 scanning electron microscopy (SEM, JEOL, Tokyo, Japan) were performed to characterize the crystal structures and morphologies of the ceramics, respectively. The Raman spectra of bulk ceramics were measured using a DXR2 laser confocal microscopy Raman spectrometer (Raman, Thermo Fisher, Waltham, MA, USA) with an argon-ion laser excitation line of 532 nm and a laser power of 1 mW. The piezoelectric coefficient *d*_33_ was measured using a quasi-static *d*_33_ tester (ZJ-6A, Chinese Academy of Science, Beijing, China). The relative dielectric permittivity *ε*_33/_*ε*_0_, dielectric loss tangent tan*δ*, and the resonance and anti-resonance frequencies of the ceramics were acquired using a precise impedance analyzer (HP4294A, Agilent, Santa Clara, CA, USA). The temperature-dependent dielectric characteristics were measured by a high-temperature dielectric measurement system (DMS-1000, Partulab, Wuhan, China). The electric hysteresis loop and field-induced strain of the ceramics were also investigated using the Radiant Precision Premier ferroelectric material test system (LCⅡ, Radiant, Redmond, WA, USA).

## 3. Results and Discussion

### 3.1. Phase Structure

The XRD patterns for Component A, Component B, (1 − *w*)PMNT-*w*PSMNT (*w* = 0.3, 0.4, 0.5, 0.6) ceramics, and the reference sample are shown in Figure 2a. As expected, the pyrochlore phase Pb_2_Nb_2_O_7_ (PDF card No. 40-0828) was formed with the featured diffraction peaks at 14.32° and 30.36° in the reference sample. Component A exhibited a pure perovskite tetragonal phase structure with no obvious impurities. Component B exhibited a rhombohedral phase structure; however, a traceable pyrochlore phase with a rhombohedral structure in Component B was detected. And, with increasing *w*, the pyrochlore phase first decreased and then increased, indicating the challenge in eliminating the pyrochlore phase, even though the contents were still far below that of the reference sample. The aggregation of Sm is the main reason for the formation of the pyrochlore phase. The formation of the pyrochlore phase can be inhibited by pre-sintering the mixed oxide of Sm, Mg, Nb, and Ti to fully diffuse Sm^3+^, and the main crystal phase could be formed by pre-sintering the mixture with PbO. In this study, niobium oxide, magnesium oxide, titanium oxide, and samarium oxide were pre-sintered at 880 °C for 4 h, promoting the homogenization of the system and minimizing the pyrochlore phase induced by the local excessive samarium concentration [18]. The ceramics prepared by sintering the mixture of two different crystalline phases displayed a higher crystal purity (99.62–99.89% perovskite structure), which is much better than the ceramics prepared by the conventional sintering method (98.64% perovskite structure). 

By fitting the diffraction peaks of the (1 − *w*)PMNT-*w*PSMNT at 44–46°, Figure 2b shows that the overlapping degree of the diffraction peaks increased gradually with *w*. With the increase in *w*, the tetragonal phase Component A decreased and the rhombohedral phase Component B increased such that the ceramics would gradually transform from the tetragonal phase to the rhombohedral phase. Consequently, double diffraction peaks of (002) and (200) transformed into a single peak of (200). When *w* = 0.6, the (002) and (200) diffraction peaks overlapped completely. Rietveld refinement was performed for (1 − *w*)PMNT-*w*PSMNT (*w* = 0.3, 0.4, 0.5, 0.6) ceramics by GSAS software (4262) [28] and the results are shown in Figure 2c–f. Compared with Component A, the symmetry of the diffraction peaks of the (1 − *w*)PMNT-*w*PSMNT ceramics became worse in the tetragonal phase (111) at 2*θ* = 31.44°. To avoid excessive grain growth, the sintering time at 1250 °C was only 1.5 h such that Component A with the tetragonal phase and Component B with the rhombohedral phase cannot be completely homogenized, resulting in the distortion of diffraction peaks to some extent [18,26]. The lattice parameters, cell volume, and density of Component A, Component B, and (1 − *w*)PMNT-*w*PSMNT ceramics are shown in Table 1. It can be seen that with the increase in *w*, the cell parameter *a* (=*b* = *c*) of the rhombohedral phase increased gradually, and the cell parameter *c* of the tetragonal phase decreased gradually. The phenomenon could be explained as follows: (1) the chemical composition of crystal particles had a gradient change when the ceramic samples with identical chemical compositions and different mixing ratios were prepared by sintering the mixture of two different crystalline phases, resulting in the change in cell parameters; (2) the cell parameters *a* and *b* of the ceramics with more pyrochlore content (*w* = 0.3 and *w* = 0.6) were relatively large, which indicates that the generation of the pyrochlore phase would alter the chemical composition of the main phase, resulting in a change in the crystal lattice. In addition, the *c*/*a* ratios of the (1 − *w*)PMNT-*w*PSMNT (*w* = 0.3, 0.4, 0.5, 0.6) ceramics were 1.0035, 1.0032, 1.0027, and 1.0017, respectively. It can be seen that the *c*/*a* of the ceramics was closer to 1 with a higher *w* because the phase structure of the ceramics tended to the cubic phase, which agreed well with the overlapping degree of diffraction peaks in Figure 2b. Compared with Sm-doped Pb(Zr*_x_*Ti_1−*x*_)O_3_ (Sm-PZT) ceramics, the *c*/*a* of Sm-PMN-PT ceramics was much closer to 1 [26,29], such that the internal stress caused by domain switching was smaller, the octahedral gap formed by oxygen atoms was relatively loose, and the micro-displacement of B-site ions was relatively easy.

### 3.2. Raman Spectra

The Raman peaks of lead-based ferroelectric materials with the ABO_3_ perovskite structure are divided into three categories: the Raman mode with a wavenumber less than 150 cm^−1^ belongs to the Pb-BO_6_ stretching mode, the one between 150 cm^−1^ and 500 cm^−1^ is a mixture of the B-O-B bending mode and O-B-O stretching mode, and the one between 500 cm^−1^ and 800 cm^−1^ is related to the B-O-B stretching mode [30,31]. According to the lattice dynamics, group theory, and Raman studies of other ferroelectrics with the ABO_3_ structure, the number of Raman modes can be used to determine the phase structure of ferroelectrics. The rhombohedral phase (*R*3*m*) has seven Raman active modes and the tetragonal phase (*P*4*mm*) has eight Raman active modes. The room-temperature Raman spectra of the (1 − *w*)PMNT-*w*PSMNT ceramics and the deconvolution of multiple Lorentzian/Gaussian peaks are shown in Figure 3, in which the fitting results agree well with the measured Raman spectra. Among the vibrational bands, the mode located at ~800 cm^–1^ can be assigned to the stretching vibration of Nb-O-Mg and B-site cations; the band at 580 cm^–1^ originates from the oxygen bending vibration; the band 500 cm^–1^ is attributed to the stretching vibration of Nb-O-Nb; the band at 433 cm^–1^ arises from the stretching vibration of the Mg-O-Mg mode; and the most intense band at 270 cm^–1^ is contributed by B site ions acting against O stretching vibration inside the octahedron [32]. The Raman spectra of ceramics with *w* = 0.3 and *w* = 0.4 can be deconvoluted into seven Raman modes using the Lorentzian/Gaussian fitting in the wavenumber range of 180–840 cm^−1^, while the ceramics with *w* = 0.5 and *w* = 0.6 have six Raman modes, which is related to the ratio of tetragonal Component A and rhombohedral Component B in the ceramics. Based on the characteristics of the Raman spectra, the ceramics with *w* = 0.3 and *w* = 0.4 were with the tetragonal phase, while those with *w* = 0.5 and *w* = 0.6 ceramics were with the rhombohedral phase. Due to the detection limit of our Raman spectrometer, the Raman modes below 99 cm^−1^ could not be accurately evaluated.

### 3.3. Microstructure

The surface of the ceramic samples was polished and etched at 1000 °C for 1 h before the morphological characterizations. Figure 4 shows that the grain size was relatively uniform in the range of 4–6 μm with an average size of 5.18 μm. It was found that a smaller grain size induced smaller domain sizes, leading to a superior piezoelectric mechanism via the clamping effect during the polarization flip [33,34]. Moreover, the ceramic grains were well-developed with clear grain boundaries and no visible precipitates presented at the grain boundaries. To some extent, the size and distribution of grains determine the properties of crystal structures [35]. The SEM images of the fresh section of the sample show both transgranular fracture and intergranular fracture mechanisms that may contribute to the comparative bonding strength between grains and grain boundaries. Moreover, the grain boundaries are angular, revealing that no glass phase remained. The grains of the ceramics were closely packed with very few pores, which agrees with the high density (95.4–97.8%) of the ceramics listed in Table 1. The higher density and fewer pores and impurities inside the ceramics contribute to higher breakdown field strengths and thus higher electrostrictive strains especially in the high electric field region [36], and reduce the absorption of light and increase the light transmission rate [37,38].

### 3.4. Dielectric Properties

Figure 5a–d show the temperature-dependent dielectric spectra of the (1 − *w*)PMNT-*w*PSMNT ceramics. The characteristic temperatures *T*_m_ (temperature of phase transition from a low-temperature ferroelectric phase to a high-temperature paraelectric phase) of the (1 − *w*)PMNT-*w*PSMNT ceramics with *w* = 0.3, 0.4, 0.5 and 0.6 were 79 °C, 79 °C, 81 °C and 86 °C, respectively, and the corresponding peak values of permittivity were 23,789, 30,190, 30,192, and 27,656. Above *T*_m_, both the permittivity and loss tangent peaks shifted along with the temperature and frequency. The results show an obvious frequency dispersion characteristic, indicating a diffusive phase transition and relaxor ferroelectric property. The dielectric spectra of the (1 − *w*)PMNT-*w*PSMNT ceramics and the reference sample at 1 kHz are shown in Figure 5e. The characteristic temperature of the reference sample was 82.2 °C, and the corresponding permittivity peak value was 19,373. The sample prepared by sintering the mixture of two different crystalline phases exhibited significantly improved permittivity. However, the Curie temperature of the PMN-PT ceramics decreased when doped with Sm. The Pb(Zr*_x_*Ti_1−*x*_)O_3_ (PZT) ceramics (from 385 °C to 335 °C [39,40]) and Pb(Mg_1/3_Nb_2/3_)-PbZrO_3_-PbTiO_3_ (PMN-PZ-PT) ceramics (from 230 °C to 184 °C [41,42]) also had these characteristics after being doped with Sm. The Curie point decreased whereas resistivity increased with increasing oxygen deficit [43], which is also the reason for the low Curie temperature of the ceramic sample. The dielectric behavior of ferroelectrics above *T*_m_ can be fitted by the quadratic law [44] for typical relaxor ferroelectrics and normal ferroelectrics.
(1)$$\frac{1}{\varepsilon }=\frac{1}{\varepsilon_{max}}+\frac{{\left(T-T_{max}\right)}^{\gamma }}{C^{\prime }}$$
where *γ* is the degree of diffuseness, and *C*′ is a constant. *γ* approaches 2 for an ideal relaxor ferroelectric, while *γ* approaches 1 for a normal ferroelectric [45]. The dielectric behavior of the (1 − *w*)PMNT-*w*PSMNT ceramics above *T*_m_ at 1 kHz was linearly fitted according to Equation (1). An example of the sample with *w* = 0.5 is shown in Figure 5f. The relaxation indexes were 1.72, 1.74, 1.83, and 1.75, respectively, showing obvious relaxation. The dielectric loss tangent was below 2% for all samples and decreased rapidly above *T*_m_, which is related to the transition from a tetragonal ferroelectric phase to a cubic paraelectric phase.

### 3.5. Piezoelectric Properties

The piezoelectric and dielectric properties of the (1 − *w*)PMNT-*w*PSMNT ceramics are shown in Figure 6 and Table 2. It can be seen that *d*_33_, *ε*_33/_*ε*_0_, and electromechanical coupling factors *k*_p_ (planar vibration mode) and *k*_t_ (thickness vibration mode) increased along with *w* and reached the maximum when *w* = 0.5. Compared to the reference sample, *d*_33_ and *ε*_33/_*ε*_0_ of the (1 − *w*)PMNT-*w*PSMNT ceramics were greatly improved. From the viewpoints of preparation process, sintering the mixture of two different crystalline phases results in a crystalline structure with the coexistence of rhombohedral and tetragonal phases, similar to the MPB, which provides more polarization directions and facilitates the reduction of domain wall energies, thus favoring the domain flipping and offering good piezoelectric performance [39,46]. In addition, a mixture of two different crystalline phases significantly enhances the disorder degree of A-site cations (Pb^2+^, Sm^3+^) and B-site cations (Mg^2+^, Nb^5+^, Ti^4+^) of the perovskite structure, which is responsible for the formation of polar-nano regions (PNRs) [22,47]. PNRs are widely considered a key signature of relaxor–ferroelectric solid solutions, which displays ultra-high piezoelectricity [48] and enhances the corresponding device performance [49]. Moreover, the dielectric loss tangent of all ceramics varied slightly while the mechanical quality factor dropped for the (1 − *w*)PMNT-*w*PSMNT ceramics. This could be attributed to the formation of a Pb vacancy at the A-site after being doped with Sm^3+^ such that the directional activation energy of the domain decreases. As the domain walls in the grains move easily, the coercive field reduces and makes the ceramics more easily polarized, thus improving the piezoelectric properties. At the same time, due to the easy movement of domain walls, the internal loss would inevitably increase, resulting in a decrease in mechanical quality factor *Q*_m_ and increase in the dielectric loss tangent [41]. 

### 3.6. Ferroelectric Properties

The hysteresis loops and field-induced strain plots of the (1 − *w*)PMNT-*w*PSMNT ceramics are shown in Figure 7. It can be seen that the loops had good rectangularity and symmetry, indicating that the materials have good ferroelectric properties. The coercive fields *E*_c_ varied slightly among ceramics, giving ~0.26 kV/mm. The remanent polarization *P*_r_ reached the maximum value of 34 μC/cm^2^ when *w* = 0.5, which is consistent with the piezoelectric and dielectric performance. Similarly, the sample with *w* = 0.5 exhibited the largest field-induced strain of 0.22%, as shown in Figure 7b. This is attributed to the equal ratio of the rhombohedral and tetragonal phases in the ceramic structure so that the domain structure exhibits a mixture of microdomains and macrodomains. It is of great significance to regulate the relationship between macrodomains and microdomains to enhance the performance of ceramics [50]. The butterfly curve is relatively thin, which indicates the small field-induced strain hysteresis, good repeatability, and fast response of the (1 − *w*)PMNT-*w*PSMNT ceramics [51]. This is very beneficial to the application of large displacement devices, such as atomic force microscopy, long-distance laser ranging calibration, submarine passive sonar, large displacement jacquard driver [52], and so on. The proposed method could precisely control the ratio of the two crystalline phases, and thus enhance the properties of the ceramics. Nevertheless, the process of this method is complex with a long experimental period, which needs to be modified for commercialization, e.g., to simplify the process for Component B. Moreover, as the low relative dielectric permittivity and high dielectric loss tangent of the materials are not suitable for miniature high-frequency devices [53], the formulation design and performance of the ceramics need to be further refined.

## 4. Conclusions

The (1 − *w*)[Pb(Mg_1/3_Nb_2/3_)_0.67_Ti_0.33_O_3_]-*w*[Pb_1−1.5*x*_Sm*_x_*(Mg_1/3_Nb_2/3_)*_y_*Ti_1−*y*_O_3_] piezoelectric ceramics with identical stoichiometric chemical compositions and pure perovskite structures were successfully prepared by sintering the mixture of two different crystalline phases. The effects of mixing ratio *w* on the crystal structure, micromorphology, and piezoelectric, dielectric, and ferroelectric properties of ceramics were studied. The results showed that sintering the mixture of two different crystalline phases allows for precise control of the crystalline phase ratio, resulting in improved piezoelectric and dielectric properties. The electrical properties varied with *w* of Component A and Component B. When *w* = 0.5, the piezoelectric and dielectric properties reached the optimal level in which *d*_33_ = 1103 pC/N, *k*_p_ = 0.66, and *ε*_33/_*ε*_0_ = 9154. This process could also be applied to the synthesis of other series of piezoelectric ceramics such as Pb(Zr_0.5_Ti_0.5_)O_3_-based ceramics, Pb(Ni_1/3_Nb_2/3_)O_3_-based ceramics, (Bi_0.5_Na_0.5_)TiO_3_-based ceramics, and K_0.5_Na_0.5_(NbO_3_)-based ceramics. The proposed method is promising but complex, so process simplification will be one of the future works to enhance the repeatability and show the potential of mass production. Moreover, as the relative clamped permittivity of the material is related to the configuration of high-frequency devices [54,55], the correlation between compositions and clamped properties of ceramics will be studied. 

## Figures and Tables

**Figure 1 materials-16-06781-f001:**
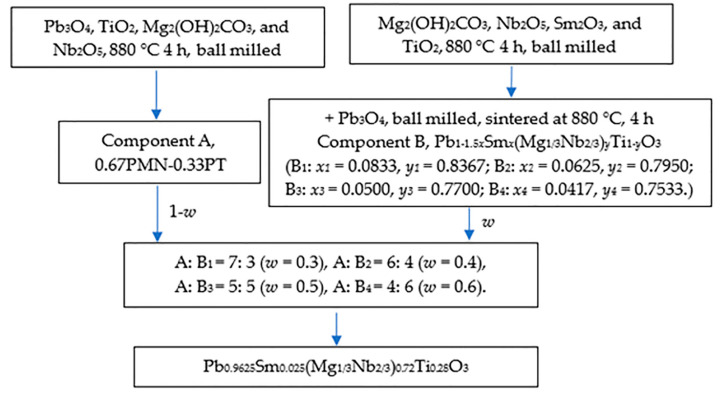
Preparation process of (1 − *w*)PMNT-*w*PSMNT ceramics.

**Figure 2 materials-16-06781-f002:**
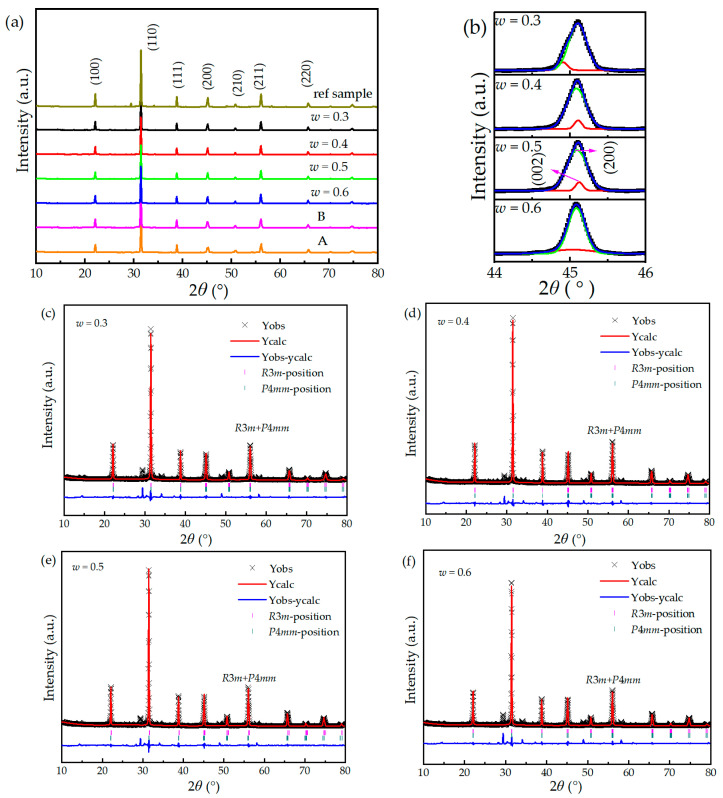
XRD patterns of (**a**) Component A, Component B, (1 − *w*)PMNT-*w*PSMNT (*w* = 0.3, 0.4, 0.5, 0.6) ceramics, and reference sample (the crystallographic index was calibrated according to the tetragonal phase). (**b**) Fitting peaks of (1 − *w*)PMNT-*w*PSMNT ceramics at 2*θ* = 44–46°. Rietveld refinement results of (**c**) *w* = 0.3, (**d**) *w* = 0.4, (**e**) *w* = 0.5, (**f**) *w* = 0.6.

**Figure 3 materials-16-06781-f003:**
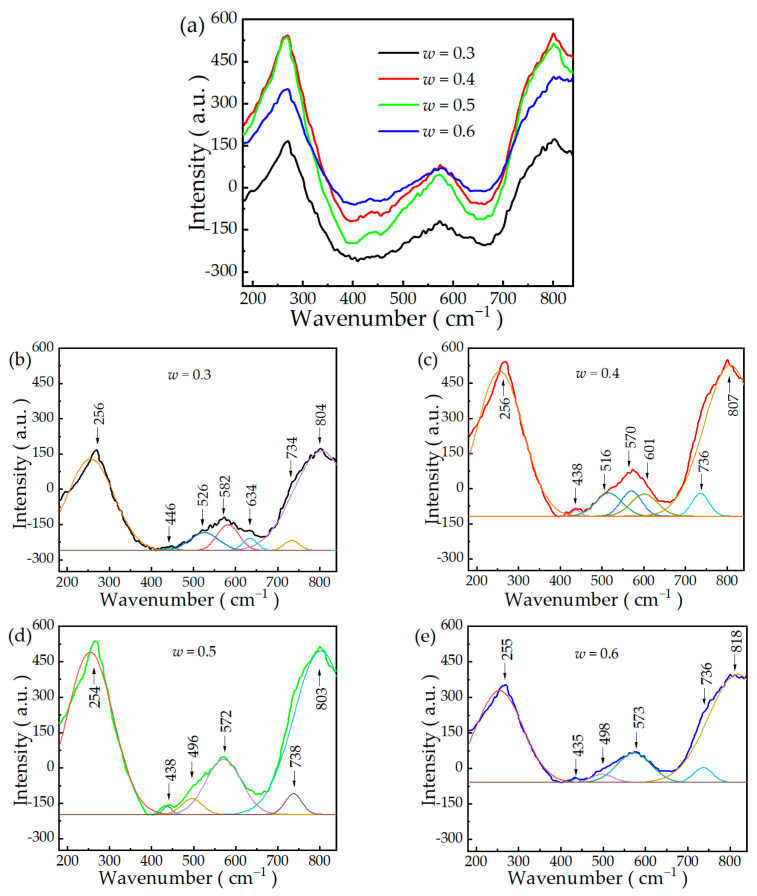
Raman spectra of (**a**) (1 − *w*)PMNT-*w*PSMNT (*w* = 0.3, 0.4, 0.5, 0.6) ceramics at room temperature; Deconvolution of multiple Lorentzian/Gaussian peaks at 180–840 cm^−1^ of (**b**) *w* = 0.3, (**c**) *w* = 0.4, (**d**) *w* = 0.5, (**e**) *w* = 0.6.

**Figure 4 materials-16-06781-f004:**
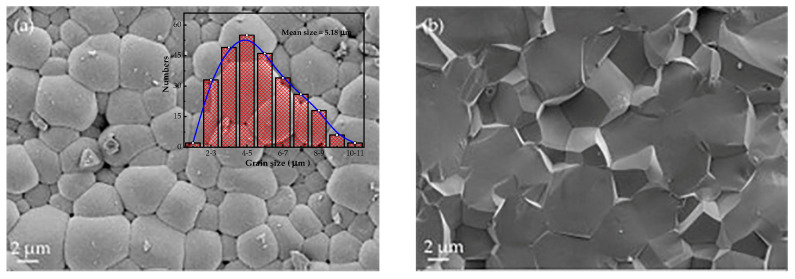
SEM images and grain size statistics chart of the (**a**) surface and (**b**) cross-section of the *w* = 0.5 ceramics.

**Figure 5 materials-16-06781-f005:**
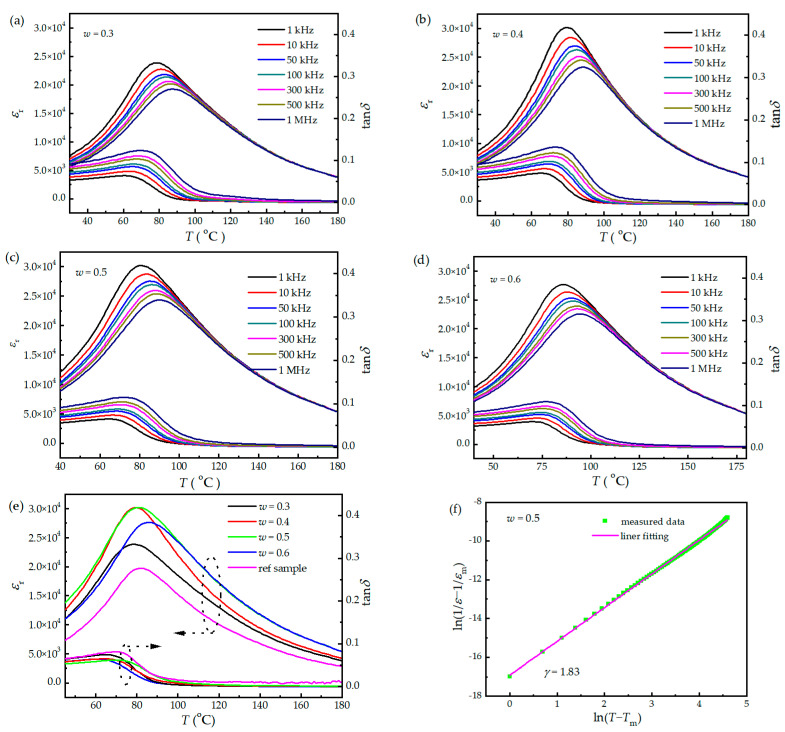
Temperature-dependent dielectric spectra of (1 − *w*)PMNT-*w*PSMNT ceramics with (**a**) *w* = 0.3, (**b**) *w* = 0.4, (**c**) *w* = 0.5, and (**d**) *w* = 0.6. (**e**) Temperature-dependent dielectric spectra of (1 − *w*)PMNT-*w*PSMNT ceramics and the reference sample at 1 kHz. (**f**) Linear fitting of dielectric behavior of (1 − *w*)PMNT-*w*PSMNT ceramics with *w* = 0.5 at 1 kHz according to Equation (1).

**Figure 6 materials-16-06781-f006:**
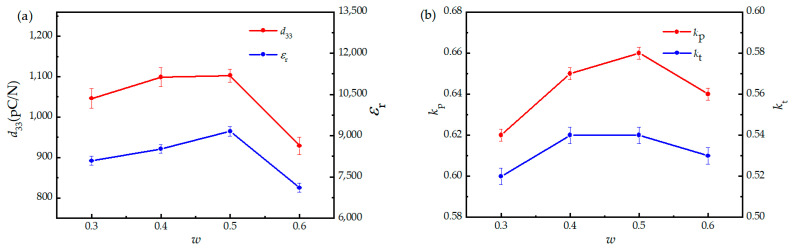
(**a**) Piezoelectric coefficient *d*_33_, relative dielectric permittivity *ε*_33/_*ε*_0_, and (**b**) electromechanical coupling factors *k*_p_ and *k*_t_ of (1 − *w*)PMNT-*w*PSMNT ceramics.

**Figure 7 materials-16-06781-f007:**
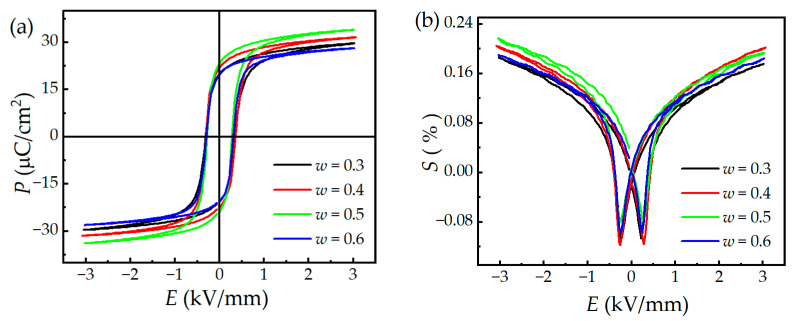
(**a**) Electric hysteresis loops and (**b**) field-induced strain plots of (1−*w*)PMN-*w*PSMN-PT ceramics.

**Table 1 materials-16-06781-t001:** Lattice parameters, cell volume, and density of Component A, Component B, and (1 − *w*)PMNT-*w*PSMNT ceramics.

Sample	Lattice Parameters (Å) Phase 2, *P*4*mm*	Lattice Parameters (Å) Phase 2, *R*3*m*	Weight Fraction (%)	Fitting Parameters	Density (%)
*w* = 0.3	*a* = *b* = 4.0129, *c* = 4.0275 *α = β = γ =* 90 volume = 64.8563 *c*/*a* = 1.0036	*a* = *b* = *c* = 4.1628 *α = β = γ =* 82.5990 volume = 70.479 *c*/*a* = 1	*P*4*mm =* 75.44 *R*3*m* = 25.56	*χ*^2^ = 11.00 *wR*_p_ = 0.0866 *R*_p_ = 0.0495	96.6673
*w* = 0.4	*a* = *b* = 4.0115, *c* = 4.0250 *α = β = γ =* 90 volume = 64.7708 *c*/*a* = 1.0034	*a* = *b* = *c* = 4.0020 *α = β = γ =* 89.3080 volume = 64.0800 *c*/*a* = 1	*P*4*mm =* 64.55 *R*3*m* = 35.45	*χ*^2^ = 8.121 *wR*_p_ = 0.0822 *R*_p_ = 0.0522	96.9128
*w* = 0.5	*a* = *b* = 4.0126, *c* = 4.0243 *α = β = γ =* 90 volume = 64.7951 *c*/*a* = 1.0029	*a* = *b* = *c* = 3.9872 *α = β = γ =* 88.1280 volume = 63.290 *c*/*a* = 1	*P*4*mm =* 40.98 *R*3*m* = 59.0200	*χ*^2^ = 7.2861 *wR*_p_ = 0.0782 *R*_p_ = 0.0506	97.8874
*w* = 0.6	*a* = *b* = 4.0146, *c* = 4.0279 *α = β = γ =* 90 volume = 64.9177 *c*/*a* = 1.0033	*a* = *b* = *c* = 4.1650 *α = β = γ =* 82.7190 volume = 70.6420 *c*/*a* = 1	*P*4*mm =* 11.58 *R*3*m = 88.42*	*χ*^2^ = 11.85 *wR*_p_ = 0.0945 *R*_p_ = 0.0528	95.4639
A	*a* = *b* = 4.0143, *c* = 4.0329 *α = β = γ =* 90 volume = 64.9886 *c*/*a* = 1.0046		*P*4*mm =* 100	*χ*^2^ = 2.7817 *wR*_p_ = 0.0502 *R*_p_ = 0.0411	96.2847
B		*a* = *b* = *c* = 4.025 *α = β = γ =* 88.5180 volume = 66.488 *c*/*a* = 1	*R*3*m =* 100	*χ*^2^ = 3.9104 *wR*_p_ = 0.0590 *R*_p_ = 0.0407	95.7749

Note: the content of the pyrochlore phase is ignored.

**Table 2 materials-16-06781-t002:** Piezoelectric and dielectric properties of (1 − *w*)PMNT-*w*PSMNT ceramics and reference sample.

Sample	*d*_33_ (pC/N)	*k* _p_	*k* _t_	*ε* _33/_ *ε* _0_	tan*δ*
*w* = 0.3	1046 ± 24	0.62 ± 0.003	0.54 ± 0.004	8068 ± 166	0.039
*w* = 0.4	1099 ± 23	0.65 ± 0.002	0.53 ± 0.003	8520 ± 159	0.041
*w* = 0.5	1103 ± 16	0.66 ± 0.002	0.53 ± 0.002	9154 ± 167	0.044
*w* = 0.6	929 ± 22	0.64 ± 0.002	0.54 ± 0.002	7102 ± 165	0.037
ref	850 ± 29	0.55 ± 0.004	0.42 ± 0.003	5631 ± 216	0.041

## Data Availability

Data are contained within the article.

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
