# Peer review of "Samarium-Doped Lead Magnesium Niobate-Lead Titanate Ceramics Fabricated by Sintering the Mixture of Two Different Crystalline Phases"

_materials, 2023, doi:10.3390/ma16206781_

Round 1
Reviewer 1 Report
See attached file

Reviewer 2 Report
The paper presents very interesting research and in my opinion, it can be published after minor revision. Detailed comments are below.
1. The abbreviation system for the tested material (1-w)[Pb(Mg1/3Nb2/3)0.67Ti0.33O3]-w[Pb1-1.5xSmx(Mg1/3Nb2/3)yTi1-yO3] is not entirely clear. The title says “PMN-PSMNT”, but based on the main chemical elements it should be “PMNT-PSMNT”. Moreover, later in the work, the authors use another abbreviation "(1-w) PMN- wPSMN-PT", which introduces additional confusion. One type of abbreviation must be used.
2. Moreover, in the general formula of the compound used throughout the work, there is an inconsistency in the marking of the variable, i.e., there is “w” (Palatino font), e.g., lines 15, 25, 69, 106, 123, 141, 147, 148, 150, 155, 162, 165, 176, 178, 179, 180, 181, 192, 204, 205, 211, 213, 230, 231, 232, 237, 251, 252, 258, 259, 260, 261, 263, 2 66, 267, 269, 278, 281, 291, 298, 305, but in some places there is "w" (Times font), e.g., lines 228, 296, as well as in the scheme on the Fig.1, figure 2, figure 3, figure 5, figure 6, figure 7.
3. Minor comments:
- lines 38, 44, “macroscopically[1].”, “easily[2-4].” missing spaces.
- lines 40, 42 there is “1. 2.” the better solution will be “(1), (2)” or “(i), (ii)”.
- lines 46, 49, 52, 54, 57, 59, 62, 66, 67, 87, 88, 89, “properties[5]”, “(MPB)[6].”,
“(Mg2+ and Nb5+)[5, 7-10]).”, “materials[11-15].”, “ceramics[16]”, “crystals[17],”, “ceramics[18-23].”, “synthesis[26]”, “system[23, 27]”, “Sm2O3[28]”, “Pb3O4(Aladdin,”, “Sm2O3(Aladdin,”, “TiO2(Aladdin,”, Mg2(OH)2CO3(Aladdin,”, “Nb2O5(Aladdin,” missing spaces. - line 69 “(1-w) PMN-w PSMN-PT” unnecessary spaces in chemical formula.
- line 101 The polarizing environment should be specified (silicone oil?). Why did the authors polarize the samples at room temperature and in such a short time? Were these optimal polarization conditions? Greater polarization efficiency is achieved at higher temperatures by using a high field and longer polarization time.
- lines 106, 123, 124 unnecessary spaces in chemical formula.
- lines 136, 151, incorrect degree symbol – the symbol there is in superscript.
- lines 137, 148, 154, 170, “concentration[16].”, “software[29]”, “extent[16, 27].”, “1[22, 27]” missing spaces.
- lines 141, 148, 150, 155, 165, 176, 178, 180, 192 unnecessary spaces in chemical formula.
- Fig.2 “w” (Times font) and letters in the space group should be written in italics.
- Table 1 letters in the space group should be written in italics.
- line 184 there is “The” should be “the”.
- lines 187, 199, “mode[30, 31].”, “octahedron[32].” missing spaces.
- line 215 “The ceramic samples were polished and etched at 1000 ℃ for 1 h before the morphological characterizations”. The SEM image shows that thermal etching was not very effective (most of the grain boundaries were not revealed), and such a microstructure could be obtained without its use. Perhaps it would have been better to use chemical etching for this purpose. In addition, SEM microstructural images for all analyzed compositions should be included in the work.
- Line 229 In the “Dielectric Properties” section the results of dielectric loss tangent tests presented in Fig. 5 are not discussed.
- lines 230, 231, 237, 251, unnecessary spaces in chemical formula.
- line 231 “Tm” describe the parameter used for the first time in the work.
- lines 232, 239, 242, 243, 244, incorrect degree symbol – the symbol there is in superscript.
- lines 233, 234,239, 241, there are “dielectric permittivity”, should be only “permittivity”.
- lines 233, 239, there are “23789”, “30190”, “30192”, “27656”, “19373” but should be “23,789”, “30,190”, “30,192”, “27,656”, “19,373”, respectively.
- lines 243, 244, 246,247, 250, “℃[20, 34])”, “℃[21, 35])”, “deficit[36],”, “law[37]”, “ferroelectric[38].” missing spaces.
- Fig.5 “w” (Times font), “tan” should by written normal font, and missing space before units (kHz, MHz). Moreover, detailed calculations in Fig. 5f are not necessary.
- lines 258, 260, 261, 263, 267, 268, 269, 278, 279, unnecessary spaces in chemical formula.
- line 267 there is “dielectric losses” buy should be “dielectric loss”, and moreover the fig.5 shows the dielectric loss tangent plots.
- Fig.6 “w” (Times font), and parameter subscripts are too large.
- lines 281, 291, 296, 298, unnecessary spaces in chemical formula.
- lines 291, 294, “ceramics[39].”, driver[40],” missing spaces.
Reviewer 3 Report
The authors have conducted a comprehensive study on the fabrication and characterization of (1-w)[Pb(Mg1/3Nb2/3)0.67Ti0.33O3]-w[Pb1-1.5xSmx(Mg1/3Nb2/3)yTi1-yO3] piezoelectric ceramics, focusing on the influence of mixing ratio (w) on various properties. The research is commendable for its systematic approach and valuable contributions to the field of piezoelectric materials. The key findings, such as the optimal properties achieved at w = 0.5 (d33 = 1103 pC/N, kp = 0.66, and ε33/ε0 = 9154), are significant and well-documented.
However, it is essential to address some shortcomings:
1) Ceramic Formulation Clarity: The paper can better explain how the ceramics were prepared, especially when it comes to varying parameters like x, y, and B1, B2, B3, B4, and make it easier for readers to grasp. [In the abstract, the author says “(1-w)[Pb(Mg1/3Nb2/3)0.67Ti0.33O3]-w[Pb1-1.5xSmx(Mg1/3Nb2/3)yTi1-yO3] piezoelectric ceramics were prepared by sintering the mixture of two different crystalline phases in which two pre-sintered precursor powders were mixed and co-fired with designated ratios (w = 0.3, 0.4, 17 0.5, 0.6)”. Looks like x and y are also variables. In the ceramic preparation, they varied these and made B1, B2, B3 and B4. In the end, the combined compound is just Pb0.9625Sm0.025(Mg1/3Nb2/3)0.72Ti0.28O3. But, the discussion is mostly based on w. They can formulate/express in a better way, which can be easier to understand for the readers.
2) Detailed Limitations Discussion: More discussion on the challenges of scaling up the fabrication method for commercial use and optimizing properties for specific applications would be beneficial.
3) Broader Implications: The paper should elaborate on how its findings could enhance existing piezoelectric devices or create new ones.
4) Research Question Clarity: The introduction needs a more focused research question or hypothesis to guide the study effectively.
5) Implications of Structural Changes: Discuss the impact of structural changes on material properties resulting from XRD analysis.
6) Microstructure Discussion: Expand on how microstructure influences material properties based on the microstructure analysis.
7) Mechanism Explanation: Provide a deeper explanation of the mechanisms driving improvements in piezoelectric properties in sections 3.5 and 3.6.
8) Enhance Conclusion: In the conclusion, discuss broader implications and suggest potential future research directions for a more comprehensive wrap-up.
Round 2
Reviewer 3 Report
Can be accepted